# Artificial Intelligence in Aptamer–Target Binding Prediction

**DOI:** 10.3390/ijms22073605

**Published:** 2021-03-30

**Authors:** Zihao Chen, Long Hu, Bao-Ting Zhang, Aiping Lu, Yaofeng Wang, Yuanyuan Yu, Ge Zhang

**Affiliations:** 1School of Chinese Medicine, The Chinese University of Hong Kong, Hong Kong, China; zihaochen@cuhk.edu.hk (Z.C.); zhangbaoting@cuhk.edu.hk (B.-T.Z.); 2Law Sau Fai Institute for Advancing Translational Medicine in Bone & Joint Diseases, School of Chinese Medicine, Hong Kong Baptist University, Hong Kong, China; longhu@hkbu.edu.hk; 3Institute of Integrated Bioinformedicine and Translational Science, School of Chinese Medicine, Hong Kong Baptist University, Hong Kong, China; aipinglu@hkbu.edu.hk; 4Guangdong-Hong Kong Macao Greater Bay Area International Research Platform for Aptamer-Based Translational Medicine and Drug Discovery, Hong Kong, China; 5Centre for Regenerative Medicine and Health, Hong Kong Institute of Science & Innovation, Chinese Academy of Sciences, Hong Kong, China

**Keywords:** artificial intelligence, aptamer, SELEX, binding, structure prediction, machine learning, deep learning

## Abstract

Aptamers are short single-stranded DNA, RNA, or synthetic Xeno nucleic acids (XNA) molecules that can interact with corresponding targets with high affinity. Owing to their unique features, including low cost of production, easy chemical modification, high thermal stability, reproducibility, as well as low levels of immunogenicity and toxicity, aptamers can be used as an alternative to antibodies in diagnostics and therapeutics. Systematic evolution of ligands by exponential enrichment (SELEX), an experimental approach for aptamer screening, allows the selection and identification of in vitro aptamers with high affinity and specificity. However, the SELEX process is time consuming and characterization of the representative aptamer candidates from SELEX is rather laborious. Artificial intelligence (AI) could help to rapidly identify the potential aptamer candidates from a vast number of sequences. This review discusses the advancements of AI pipelines/methods, including structure-based and machine/deep learning-based methods, for predicting the binding ability of aptamers to targets. Structure-based methods are the most used in computer-aided drug design. For this part, we review the secondary and tertiary structure prediction methods for aptamers, molecular docking, as well as molecular dynamic simulation methods for aptamer–target binding. We also performed analysis to compare the accuracy of different secondary and tertiary structure prediction methods for aptamers. On the other hand, advanced machine-/deep-learning models have witnessed successes in predicting the binding abilities between targets and ligands in drug discovery and thus potentially offer a robust and accurate approach to predict the binding between aptamers and targets. The research utilizing machine-/deep-learning techniques for prediction of aptamer–target binding is limited currently. Therefore, perspectives for models, algorithms, and implementation strategies of machine/deep learning-based methods are discussed. This review could facilitate the development and application of high-throughput and less laborious in silico methods in aptamer selection and characterization.

## 1. Introduction

Aptamers are single-stranded nucleic acids (both DNAs and RNAs) with a high affinity toward target molecules [1,2]. A number of aptamers were selected to against a wide variety of target molecules, such as proteins and viruses [3]. Aptamers are usually referred to as “chemical antibodies” because of their high selectivity and binding affinity toward target molecules [4]. Compared to antibodies, aptamers possess the following merits. Firstly, the structures and sizes of aptamers are more flexible and smaller than those of antibodies. Thus, aptamers can recognize and bind to the targets which are inaccessible for antibodies, such as smaller targets or some hidden binding domains [3]. Secondly, aptamers are much cheaper and require less time for production than antibodies since they could be massively synthesized [5]. Last, aptamers are more stable under most conditions, which increases their shelf life [6]. Due to these predominant merits, aptamers are taken as promising competitors to antibodies in diagnostics, therapeutics, cell imaging, biosensor, biochip, and drug delivery [7,8].

Aptamers are usually identified through an in vitro experimental approach firstly implemented in the 1990s named systematic evolution of ligands by exponential enrichment (SELEX) [1,2]. SELEX has the ability to select aptamers bounding to target molecules with high selectivity and binding affinity [9]. Firstly, a random nucleic acid library that contains 10^14^–10^15^ random oligonucleotide strands is created. Secondly, the library is incubated with the target molecules to form a complex of target–oligonucleotides. Thirdly, the complex of target–oligonucleotides is separated from the rest of the unbound library pool. Then, with unbound sequences washed away, the specifically bound oligonucleotides are then eluted from targets. Finally, the amplification and new selection cycle of target-binding oligonucleotides are conducted. This whole process of high-affinity aptamers selection normally contains 6–15 rounds. However, there are some barriers for the SELEX technology, including the following: (1) it requires weeks, even months to acquire aptamer candidates; (2) the successful rates of aptamer candidates are still low; (3) only a limited number of representative aptamer candidates from the next-generation sequencing data could be synthesized for affinity characterization. 

Recently, some researchers used computational methods to select aptamer candidates because of their convenience and low cost [10,11]. These methods aim at predicting the aptamer affinity toward targets through structural information [12]. Many online servers such as RNAfold and RNAComposer have been proposed for predicting the secondary structure and three-dimensional (3D) structures of RNAs/DNAs [13]. Like other RNAs/DNAs, the structural information of aptamers could also be obtained by these online servers [12]. Similarly, molecular docking and molecular dynamics usually used in protein compounds selection by structural information have shown fitness for protein aptamers selection [14]. 

Artificial intelligence (AI) including machine/deep-learning algorithms has inspired novel computational methods for selection of aptamer candidates with high affinity and specificity to target molecules in drug discovery [15]. Some machine/deep-learning methods have been shown to outperform a wide range of classical binding affinity prediction methods such as molecular docking and virtual screening tools [16]. In the previous study, our group applied AI to develop small molecules specifically targeting miRNA–mRNA interactions by using a random forest model [17]. Although not commonly used in aptamer-based discovery currently, machine/deep-learning methods are promising in aptamer–target affinity prediction. Machine/deep-learning methods do not require the structural information of aptamers and thus are able to effectively explore much larger amounts of experimental data. Furthermore, the performance of machine/deep-learning methods could be improved with larger training datasets [16]. Therefore, with these advantages, perspectives of employing machine/deep-learning algorithms in aptamer affinity prediction are discussed in this review. 

## 2. Aptamer Affinity Prediction through Structural Information

In the past decade, computational methods of bioinformatics have been proposed to facilitate the selection of potential aptamers through the prediction of aptamer structure. The computational methods provide convenient and accurate ways to select aptamers with high affinity. A typical modeling workflow of aptamer selection by computational methods comprises four steps (Figure 1) [11]. Firstly, the secondary structures of aptamers are predicted by their sequences. Secondly, prediction and optimization of the tertiary structures are adopted by the secondary structures. Subsequently, rigid or flexible molecular docking is performed to predict the structure of the aptamer–target complex. Last, molecular dynamic simulations are performed for evaluations of the stability of the aptamer–target binding modes. 

### 2.1. Secondary Structure Prediction for Aptamers

Aptamer secondary structure, an abstract form of tertiary structure, plays a pivotal role in binding between aptamer and target molecules [18]. For example, the binding affinity could be elevated by forming secondary structures such as G-quadruplex, hairpin loop, and T-junction [19]. In addition, aptamer secondary structure contributes to the aptamer 3D structure prediction [20]. Different computational algorithms have been developed for the prediction of aptamer secondary structures (Table 1).

The computational principles are similar for DNA and RNA aptamers, though their components are different. Current online servers for secondary structure prediction can be classified into two major categories: free energy-based methods and sequence alignment-based methods [26]. RNAfold could predict the secondary structure with the minimum free energy by inputting a single sequence [30]. RNAfold was selected to predict the tetracycline aptamer [21]. Another free energy-based prediction method is Mfold [22]. The “M” in Mfold simply refers to “multiple.” In one study, four ssDNA aptamers were selected to inhibit the activity of angiotensin II, and the Mfold program was used to predict the secondary structure of the aptamers [23]. The RNAstructure online web server, a free energy minimization method which was first reported in 1998, has been expanded to contain many structure prediction methods, including maximum expected accuracy [31], stochastic sampling [32], exhaustive traceback [33], and pseudoknot prediction [34]. The secondary structures of DNA aptamers against 17β-estradiol were predicted using RNAstructure [35]. Vfold2D is a free energy-based program that predicts RNA 2D structures using the RNA motif-based loop entropies [36]. The secondary structures of aptamers against human immunodeficiency virus-1 reverse transcriptase (HIV-1 RT) were predicted from the sequence by using the Vfold2D program [36]. The CentroidFold online web server, a sequence alignment-based method, can predict common secondary structures for multiple alignments of RNA sequences by using an averaged gamma-centroid estimator [28]. In the previous study, the CentroidFold web server was used to predict the secondary structures of RNA aptamers targeting angiopoietin-2 [29].

There was no parallel comparison for the accuracy of these secondary structure prediction methods. Therefore, we randomly chose five aptamers with known secondary structures and performed a comparison analysis (Table 2). Firstly, the 3D structures of five aptamers were downloaded from the Protein Data Bank (PDB) database. Secondly, the RNApdbee server [37] aiming at extracting the RNA 2D structures from the PDB file was selected to obtain the real 2D structures of aptamers. Then, we predicted the 2D structures of the aptamers using RNAfold, RNAstructure, CentroidFold, Vfold2D, and Mfold online servers. Finally, accuracies of these servers were calculated by comparing the coincidence between the predicted and real 2D structures of the aptamers. As shown in Figure 2, basically all prediction methods have high accuracy. RNAfold and RNAstructure were the most accurate online servers to predict the 2D structures of the aptamers since they predicted the same results, and their mean accuracies (0.94) were ranked first.

### 2.2. 3D Structure Prediction for Aptamers

#### 2.2.1. Structure Prediction for RNA Aptamers

Aptamers can form complexes with their target proteins to achieve diverse biological functions. Since 3D structures determine the functions of biological molecules, the precise 3D modeling for aptamers is very important. Currently, four online web servers, RNAComposer, 3dRNA, Vfold3D, and SimRNA, primarily proposed for RNA 3D structure prediction have been used to adopt structure construction for RNA aptamers. These four online web servers could be divided into two categories, based on fragment methods (RNAComposer, 3dRNA, and Vfold3D) and energy-based methods (SimRNA) (Table 3). The input data for these online web servers include RNA sequence and RNA secondary structures in the dot-bracket notation [38]. 

In the modeling process of RNAComposer, the input secondary structure is first divided into fragments and then matched with 3D elements. Secondly, RNAComposer assembles these matched 3D elements to form a complete 3D structure. The final 3D structure of the RNA aptamer comes from the energy minimization of the complete 3D structure. Hu et al. predicted the RNA aptamers’ 3D structures using RNAComposer and then selected RNA aptamers targeting angiopoietin-2 with high binding affinity [29]. 

Another fragment-based 3D RNA structure method is 3dRNA; it employs the secondary elements, including the helix and loops [39]. It finds the 3D template for each secondary element and then assembles the template together as per the final predictions. An RNA aptamer targeting *Streptococcus agalactiae* surface protein was studied, and its 3D structure was predicted by 3dRNA [40]. 

Similarly, Vfold3D identifies the motif, such as helices and loops in the RNA 2D structure, and then finds the best templates for each motif [26]. Then, the 3D structure of these templates is assembled, and energy of the structures is minimized to construct the 3D structure of the whole RNA aptamer. In a study about selecting an aptamer targeting the prostate-specific membrane antigen, the 3D structure of the RNA aptamer was predicted by the Vfold3D online web server [41]. 

SimRNA, a computational method for RNA folding simulations, uses a coarse-grained representation of the nucleotide chain and a knowledge-based energy function to produce the most energetically favorable 3D conformations [42]. In a study about selecting the aptamers targeting angiopoietin-2, the 3D structure of the RNA aptamer was designed using the SimRNA web server [43]. 

We compared the accuracy of these 3D structure prediction methods. The process of obtaining 3D structures and secondary structures of five aptamers was described in Part 2.1. We predicted the 3D structure of the aptamers using the RNAComposer, 3dRNA, Vfold3D, and SimRNA online servers. The root-mean-square deviation (RMSD), the measure of the average distance between the atoms, was used to compare the accuracies of these servers by aligning the predicted structures and the real 3D structures of aptamers from the PDB database. As shown in Figure 3, basically all prediction methods have high accuracies for the 3D structure prediction for short aptamers (less than 40 nt) and the accuracies of RNAComposer, 3dRNA, and SimRNA were obviously reduced for long aptamers (Aptamer 5, which is 83 nt). These results suggested that the optimal length for aptamer structure prediction is less than 40 nt. Interestingly, Vfold3D was consistently accurate for all aptamer structure predictions. The number of aptamer structures used in this study was limited, and more original data are needed to further validate the accuracy of these aptamer 3D structure prediction methods. 

We also compared the variation of binding energies between the predicted structures and the determined structures of aptamers in docking with the target proteins. Aptamer 2LUN and its target protein (*B. anthracis* ribosomal protein S8) was selected as the reference group. The 3D structures predicted by Vfold3D, SimRNA, RNAcomposer, and 3dRNA were used to check the variation of binding energies to the determined structure of the aptamer. The 3D structure of the target protein was downloaded from the PDB database (ID 4PDB), and the binding sites of the protein to the aptamer were set as LYS54, GLN80, ALA114, SER130, and GLY147. The molecular docking was completed using ZDock (Discovery Studio), which could calculate the binding energy values with the following equation: binding_energy = complex_energy–(protein_energy + ligand_energy). As shown in Table 4, the aptamer predicted by the Vfold3D website had the lowest binding energy variation. Since the Vfold3D was consistently accurate in all aptamer structure predictions and had the smallest variation to the determined structure of the aptamer in molecular docking, Vfold3D is recommended to be used for the aptamer 3D structure prediction. 

#### 2.2.2. Structure Prediction for DNA Aptamers

Although DNA aptamers have been widely used in biomedical applications, the computational methods for predicting 3D structures of DNAs are fewer than their RNA counterparts [20]. 3D structure prediction methods for RNAs are commonly used for DNA structure prediction. RNAComposer could be used to generate 3D structures of RNAs and then transformed to DNA structures [44,45]. For example, Iman et al. introduced a workflow for predicting 3D structures for DNA aptamers [20]. The workflow could be divided into four main steps. Firstly, the Mfold online web server was used to predict the secondary structure of DNA aptamers. Secondly, the Assemble2/Chimera software was used to construct 3D RNAs. Thirdly, the VMD software was used to translate 3D RNA structures to 3D DNA structures. Finally, the VMD software was used to refine 3D structures of DNA aptamers. The validation of the workflow was conducted in a pool of 24 DNA molecules and aptamers with available 3D structures. The validation results indicated that the predicted structures of DNA molecules were in good agreement with the real 3D structures.

### 2.3. Docking

Molecular docking is a crucial tool to predict the predominant binding mode(s) and binding sites of the protein and the ligand. For molecular docking tools, there are two main steps: firstly, searching all potential binding poses between the protein and the ligand; secondly, providing a scoring function to evaluate these binding poses [46]. Among these molecular docking tools, ZDOCK, MDockPP, AutoDock, AutoDock Vina, and DOCK have shown successful results in aptamer design. 

The ZDOCK program uses the fast Fourier transform (FFT) algorithm to search and obtain all the binding poses and utilizes a combination of shape/electrostatics to score these binding poses [47]. ZDOCK achieved a high predictive accuracy on protein–protein docking benchmarks, with a > 70% successful rate in the top 1000 predictions [48]. Computational simulations of Ang2–aptamer interactions were performed by using the ZDOCK and ZRANK docking functions in Discovery Studio 3.5 [29]. Another FFT-based docking algorithm is MDockPP [49]. MDockPP globally samples all putative binding poses, and then the binding poses are refined with a knowledge-based scoring function. Validation results demonstrated that MDockPP correctly modeled for six out of 11 targets. In a study about designing the aptamer targeting the prostate-specific membrane antigen (PSMA), the molecular docking was completed by MDockPP [25]. 

DOCK uses a shape-matching approach to sample alternative binding poses [50], and the binding poses are scored using the Assisted Model Building with Energy Refinement (AMBER) molecular force field [51]. In a study for identifying cytochrome p450 aptamers, a series of aptamers were designed using DOCK [52]. AutoDock is a suite of software for molecular docking and contains two applications, AutoDock4 and AutoDock Vina [53]. AutoDock4 calculates the free energy to score binding poses, while AutoDock Vina uses an empirical scoring function to score the binding poses [54]. AutoDock4 performs better in more hydrophobic, poorly polar, and poorly charged pockets, while AutoDock Vina is more successful in polar and charged binding pockets [55]. In the research about designing anti-Ang2 aptamers, the molecular docking process was completed by AutoDock Vina [43].

### 2.4. Molecular Dynamics (MD)

After the molecular docking, MD simulations need to be performed to evaluate the stability of protein–aptamer complexes and determine the binding energies [11]. The typical MD process contains the initial molecular configuration describing the atomic interactions and model physics, running a simulation, and recording observations from the trajectory [56]. Such simulations evaluate millions of interactions of particles for billions of time steps, which can require extraordinary amounts of computational power and time. Currently, MD are available in many software packages, such as AMBER [57] and GROMACS [58]. The binding energy of protein–aptamer complexes could be simply calculated by subtracting the sum of protein energy and ligand energy from the complex energy [59]. Shcherbinin et al. investigated and designed aptamers toward cytochrome p450 [52]. The GROMACS 4.0 program was used to perform MD simulations for DNA aptamers against human thrombin [60].

### 2.5. Structure Prediction of G-Quadruplex (G4) Aptamers

G4 are noncanonical nucleic acid structures formed by particular guanine-rich oligonucleotides [61]. The main G4 component is the guanine tetrad, a cyclic planar arrangement of four guanines associated through Hoogsteen hydrogen bonds. Besides, the cations in the center of the G4 could further stabilize the G4 structure [62]. Guanine-rich aptamers have the ability to fold into stable G4 structures under physiological conditions and recognize different proteins [63]. The advantages of G4-structured aptamers contain thermodynamical and chemical stability, low immunogenicity, and resistance to serum nucleases [64]. G4-structured aptamers have been used as therapeutic and diagnostic tools, such as anticoagulants [65] and anticancer agents [64]. 

Structure determation tools such as NMR [66] and X-ray crystallography [67] have been used to characterize G4 structures. These techniques are not suitable for scanning multiple G4 structures, but in silico methods could identify G4 structures on a whole-genome scale [68]. Puig et al. systematically summarized the computational methods for G4 formation detection from DNA/RNA sequences. The G4 structure detection methods contain the regular expression matching approaches, scoring approaches, sliding window algorithms, and machine-learning models. Among these computational methods, qsfinder (scoring approach) outperformed all the other prediction tools [68]. 

Apart from DNA/RNA sequences, the binding affinities of aptamers to their targets could be influenced by buffered solutions and presence of other aptamers. For example, the Tris/K^+^ buffer could favor the G4 structures formation and increase the affinity between the aptamer and the target. On the other hand, the PBS/Mg^2+^ buffer could destabilize the G4 structure and is unfavorable for the binding of aptamers to target proteins [69]. Metal ions control the folding of G4 and are critical to the inhibitory activities of aptamers [70]. Interestingly, Troisi et al. showed that the binding of a G4 aptamer at one exosite to thrombin increases the binding affinity of another aptamer to thrombin at a different exosite [71]. 

## 3. Aptamer Affinity Prediction through Machine/Deep Learning

Structure-based methods are not suitable or capable of scanning and predicting the affinity of a vast number of sequences to one target at the same time. The machine-/deep-learning methods could be directly and efficiently used for prediction of massive sequences from the next-generation sequencing data. In addition, machine-/deep-learning methods could provide more accurate affinity prediction. 

Machine learning consists in extracting knowledge from data and determining the internal relationships that can improve themselves without human intervention [72]. Deep learning is one of the machine-learning techniques and imitates the human brain with deep networks capable of learning and analyzing data [73]. Both deep learning and machine learning are subsets of artificial intelligence. To the best of our knowledge, some studies have been done to identify high binding affinity aptamers by machine learning and deep learning [10,74].

### 3.1. Machine Learning in Aptamer Prediction

Machine learning (ML) methods can be divided into feature-based ones and similarity-based ones. The feature-based methods use descriptors to generate feature vectors while the similarity-based methods use the “guilt by association” rule [15]. The binding affinity between the aptamer candidates and their targets are predicted based on the similarities between the candidates. The similarities are commonly evaluated by clustering analysis via sequence- or structure-based features.

#### 3.1.1. Sequence-Based Clustering

Sequence clustering tools discover the closely related sequences by identifying similarities between the actual sequences (A/T/G/Cs) of different aptamers in a SELEX pool. These methods run fast since they treat aptamers as simple sequence strings, and therefore leverage previously developed highly efficient string comparison algorithms. AptaCluster can calculate the similarities between aptamer sequences based on the local sensitive hashing (LSH) method, which can compare sequences with a reduced number of dimensions [75]. FASTAptamer and PATTERNITY-Seq both use the Levenshtein distance to cluster sequences [10,76]. The Levenshtein distance is determined by calculating the minimum number of insertions/deletions/substitutions needed to convert one word into another. By only using strings of A/T/G/Cs to represent aptamers, these sequence clustering models are able to achieve a high speed of analyzing large SELEX datasets. However, the accuracy is a drawback of these models since structural information which is critical to determine the affinity of an aptamer’s binding is not considered.

#### 3.1.2. Structure-Based Clustering

Structure-based clustering models attempt to cluster aptamers based on shared structural motifs and information and predict the binding affinity based on similarity to the aptamers with known affinity to targets. AptaTrace and APTANI are two models both clustering aptamers based on their structure motifs [77,78]. AptaTrace tries to associate each structural motif observed in a library of aptamers with its impact on enrichment levels. What is more, in each selection round, a specific structure can be predicted for each aptamer, and the candidates are subsequently ranked by structural enrichment. APTANI is a tool based on the AptaMotif algorithm to analyze SELEX data [79]. AptaMotif is an ensemble-based method to extract structure motifs efficiently from SELEX-derived aptamers. SMART-Aptamer was developed to identify high binding affinity aptamers by multilevel structural analysis and unsupervised machine learning [80]. This model uses both motif finding and cluster-based strategies while considering the overall secondary structure. The RaptRanker uses clustering, scoring, and ranking methods to identify aptamers with high binding affinity [81]. Firstly, the unique sequences in the dataset are determined, and the nucleotide sequence and secondary structure features are used to cluster all subsequences of the unique sequences. Then, to identify aptamers with high binding affinity, the average motif enrichment (AME) score is applied to each unique sequence and calculated based on the frequency of subsequence clusters. These models can incorporate domain knowledge and capture structural information about aptamer binding; they tend to take significantly longer time to run since they need to predict secondary structures. Additionally, these tools are based on clustering which may be biased towards aptamers that are highly similar to the already observed sequences. Therefore, these models limit our ability to optimize SELEX results.

#### 3.1.3. Feature-Based Machine Learning

Supervised machine learning consists in learning a function from labeled training data, and this function can predict outcomes for unlabeled data [82]. There have been some studies to predict the aptamer binding ability by supervised machine learning. Li’s team proposed a method to integrate the features derived from both aptamers and their target proteins in the Aptamer Base [83,84]. They used the maximum relevance minimum redundancy (mRMR) method and the incremental feature selection (IFS) method to select the features, and then a random forest model was developed. Aptamers against corresponding targets including human interleukin 17A, prothrombin, and human toll-like receptor 3 ectodomain were used to test the accuracy of this method. Zhu et al. reported an ensemble strategy to predict aptamer–protein interaction based on sequence characteristics derived from aptamers and the target proteins in the same dataset as in Li’s study [85]. A sparse autoencoder was applied to characterize features for target protein sequences. Then, gradient boosting decision tree (GBDT) and incremental feature selection (IFS) methods were applied to obtain the optimum combination of sequence characters. Eventually, a prediction model was constructed based on three sub-support vector machine (SVM) classifiers. Nevertheless, these models are empirical and knowledge-based and require extensive training. Moreover, these shallow machine learning models with sequence data usually cannot fully learn key characteristics (such as distance correlation), which leads to inaccurate prediction.

### 3.2. Deep Learning in Aptamer Prediction

Deep-learning models may have better performance than machine-learning models because they learn the features without the requirement of feature engineering, thus can model long-range and multi-body atomic interactions [74]. The representation of input data and the deep-learning architecture are two main aspects in deep-learning applications. According to the input data, the current studies can be divided into sequence-based and structure-based models to predict the aptamer–target binding affinity [15]. Meanwhile, the widely used deep-learning architecture in aptamer studies is based on the recurrent neural network (RNN), the convolutional neural network (CNN), or the general regression neural network (GRNN). RNN-based models are commonly used with aptamer sequences since they can process the sequenced information as inputs in deep networks [86]. The GRNN is slightly different from radial basis neural networks and the method is that every training sample represents a mean to a radial basis neuron [87]. CNN-based models can train and test input data through a series of convolution layers with filters, pooling and fully connecting layers, thus they are often used to predict the binding ability based on structural information [88]. 

Despite the power and accuracy of deep learning models to predict the aptamer binding affinity, few applications have been reported so far. Michael et al. predicted aptamer binding affinity by applying the conditional variational auto encoder (CVAE) model for aptamers against a small molecule daunomycin [74]. The CAVE model used a bidirectional long short-term memory network (LSTM), an RNN-based method, as the encoder and a series of parallel feedforward networks as its decoder. This model can capture the complex relationship of aptamer sequences to predict the novel aptamer sequence with high affinity without inferring the structural data. In addition, Yu et al. developed quantitative structure–activity relationships (QSAR) based on the GRNN to predict the binding affinity between aptamers and the influenza virus [89]. Molecular descriptors were calculated via the GRNN model for extracting the structural features from aptamer sequences. These studies demonstrated the feasibility to calculate the aptamer binding affinity by deep-learning models and to predict novel aptamer candidates with higher affinity.

## 4. Perspectives

Here, we attempted to suggest some possible future avenues to improve prediction methods by several subsections based on the different aspects of improvements towards prediction of aptamer binding affinity.

### 4.1. Machine/Deep Learning in Aptamer 2D Structure Prediction

Recently, machine-learning methods such as KNetfold and SPOT-RNA aiming at predicting secondary structures of RNA have been considered to provide a novel method for optimizing the predicted secondary structures of aptamers. KNetfold, a hierarchical network of k-nearest neighbor classifiers, uses RNA sequence alignment to predict a consensus RNA secondary structure [90]. KNetfold showed a significant improvement compared with the secondary structure prediction methods PFOLD and RNAalifold [90]. SPOT-RNA is built by two-dimensional deep neural networks and transfer learning [91]. Initially, models of ResNets and LSTM networks are trained in a bpRNA dataset which contains more than 10,000 nonredundant sets of RNA sequences with annotated secondary structure from bpRNA at the 80% sequence identity cutoff [92]. Then, the models obtained from the bpRNA dataset are transferred to further train on base pairs derived from high-resolution nonredundant RNA structures (less than 250). SPOT-RNA, with a freely available server and standalone software, improved by around 10% in the Matthews Correlation Coefficient (MCC) and F1 score over the next best program by comparison [91]. 

### 4.2. Machine/Deep Learning in Aptamer 3D Structure Prediction

For protein design problems, progress in the field of deep generative models has spawned a range of promising approaches such as AlphaFold (the AlphaFold team). It could predict 3D structures of proteins with high accuracy, even for proteins with fewer homologous sequences. The prediction process of AlphaFold consists in (1) making predictions of the distances between pairs of residues; (2) constructing a potential of mean force by residue distances to describe the shape of a protein. 

On the other hand, with the development of computational methods, some researchers tried to apply deep-learning methods to achieve 3D genome folding from DNA sequences which may inspire the optimization of 3D structure prediction for DNA. Akita could predict genome folding from DNA sequences with a deep convolutional neural network [93]. Compared to the previous machine-learning approaches, Akita could also predict the effects of DNA variants and the derived features of genome folding. Akita consists of the “trunk” based on Baseji and the “head”, whose functions include learning DNA motifs with combined grammar sin genome folding and recognizing the features relationships. Akita could only represent DNA genome folding and it is not enough to predict the details of 3D structures for DNA aptamers. However, this method showed the potential of deep learning in 3D structural prediction of DNA aptamers.

### 4.3. Improvement and Potential of Machine/Deep Learning in Aptamer Prediction 

For machine learning, there have been some models applied in predicting binding affinity of small-molecule drugs, which can be great references for predicting aptamer affinity. The Kronecker regularized least squares (KronRLS) and SimBoost approaches were developed to achieve this goal; both are based on the hypothesis that similar drugs tend to have similar targets [94,95]. KronRLS’s features include different types of drug–drug and protein–protein similarity score matrices defined through the Kronecker product of drugs and targets. SimBoost is a non-linear method that uses gradient boosting regression trees to predict drug–target binding affinity. Similarity matrices and constructed features are both used in this model. Comparing to the simple clustering methods, KronRLS can better reflect the true complexity of the drug–target prediction problem in practical applications since it is based on a more realistic formulation of the drug–target prediction problem. The regularized least squares approach (RLS) has been used in many applications [96]. SimBoost overcomes the limitation of obtaining only linear dependencies of drug–target binding. Furthermore, SimBoost applies a confidence score to a prediction because of the bias in the training datasets. In aptamer binding affinity prediction, we can also apply the RLS model or gradient boosting regression trees to build the prediction model. Ashtawy et al. proposed a machine learning-based score function to predict Drug-Target Binding Affinity (DTBA) between drugs and targets [97]. This score function utilizes a total of six regression methods, including multiple linear regression (MLR), multivariate adaptive regression splines (MARS), k-nearest neighbors (kNN), support vector machines (SVM), random forests, and boosted regression trees (BRT). They use the training dataset in collaboration with cross-validation to obtain appropriate values of the parameters because most of these machine-learning methods benefit from parameter adjustments before their use in prediction. This model can outperform other models with the best or near-best performance in most datasets because it integrates several machine-learning models. For prediction of aptamer binding affinity, the model can get more information from the training data, find true interrelationships, and provide high prediction accuracy if we could combine several machine-learning models.

For deep learning, here, we summarized several concepts of deep-learning models applied in small-molecule drug discovery, which may improve the accuracy and robustness in current aptamer studies: (a) CNN-based models to extract structural information; In the Pafnucy algorithm, a CNN model was applied to extract the input drug structure with 3D grids and 4D tensors [98]; 3D-CNN models have been used to build multiple AK-Score predictors with a better performance than a single predictor [99]; (b) artificial neural network (ANN) models to combine multiple models; for the BgN-Score (ensemble neural networks through bagging) and the BsN-Score (ensemble neural networks through boosting), total features are ensembled from each protein–ligand complex and then boosted in a stage-wise fitting manner with ANN learners [100]; it has a significant improvement with ANN comparing to a single neural network; furthermore, ANN-based ensembling approach does not need to further modify network architectures and can be easily applied to most existing models; (c) combination of RNN-based and CNN-based models; in the DeepAffinity algorithm [101], input protein sequences are encoded by an RNN-based Seq2seq model, which has achieved great success in modeling sequence data in natural language processing [102]; then, the decoded features are trained in a CNN-based model; attention mechanisms are also jointly applied to focus on a certain part, compound, or protein; (d) the word-based CNN model to represent the sequence information: word-based methods use a set of words to represent the sequence and can detect shorter lengths of residues that represent protein characteristics, and thus this aspect is considered as an advantage comparing to character-based methods; DeepDTA and WideDTA both use CNN to form prediction models for drug–target binding affinity [103,104]; in the input representation part, DeepDTA is character-based, while WideDTA is word-based; furthermore, WideDTA incorporates four text-based information sources: protein sequence, ligand sequence, motif and domain sequences for proteins, and the maximum common substructures (MCS) sequence for the ligand, which provides more information than the focus only on the protein–ligand sequence; as a result, WideDTA features a better performance than DeepDTA; (e) generative adversarial networks (GANs)-based methods to improve the performance of a large database: GANsDTA, a semi-supervised GANs-based method [105], generates fake samples according to a given noise distribution. Then, all the fake and real samples are input to the discriminator to train a local feature extractor for better classification. As a result, a GANs-based model can achieve similar performance to DeepDTA and it may achieve better prediction for large datasets. Diagrams of four popular deep-learning models (ANN, GAN, RNN-based LSTM, and CNN) are shown in Figure 4, as an illustration of different learning processes among these algorithms. These deep-learning models all use neural networks to learn from large data, which is inspired by the human brain. They all have the input layer and the output layer. Figure 4a is an artificial neural network (ANN), which consists of the input layer, hidden layers, and the output layer. Every node in one layer is connected to other nodes in the next layer. Figure 4b is a generative adversarial network (GAN), which has two main components: a generator model for generating new data and a discriminator model for classifying whether the input data are real, come from the domain, or fake, generated by the generator model. Figure 4c is an RNN-based long short-term memory networks (LSTM), which consists of the input gate, the output gate, and the forget gate. LSTMs take inputs from the previous timestep into account when modifying the model’s memory and input weights. Figure 4d is a convolutional neural network (CNN), which consists of convolution layers with filters, pooling layers, fully connected layers (FC), and softmax functions. CNNs are often used in image research.

## 5. Conclusions

Structure-based methods are the most used in computer-aided drug design. Advanced machine-/deep-learning models have witnessed successes in predicting the binding abilities between targets and ligands in drug discovery and thus potentially offer a robust and accurate approach to predict the binding between aptamers and targets. More models could be applied for machine/deep learning for aptamer binding prediction. Moreover, a combination of structure-based and machine-/deep-learning methods could be a promising approach in aptamer–target binding ability prediction. This review could facilitate the development and application of high-throughput and less laborious in silico methods for aptamer selection and characterization.

## Figures and Tables

**Figure 1 ijms-22-03605-f001:**
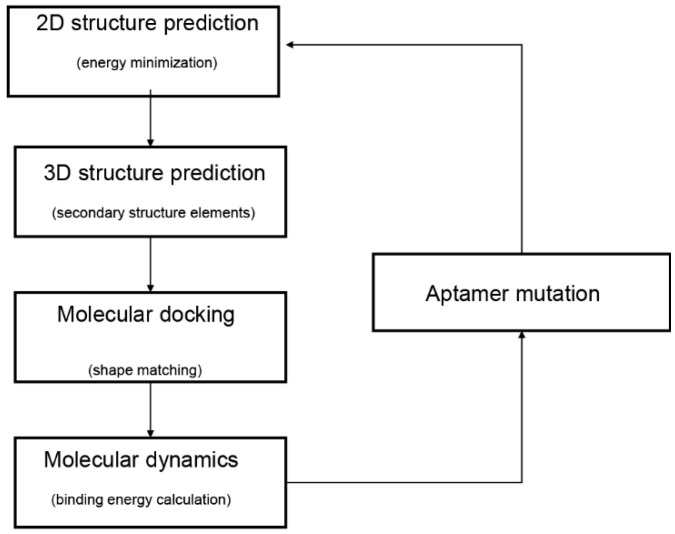
Typical workflow of in silico aptamer design and analysis. The diagram was adapted from Buglak et al. [11].

**Figure 2 ijms-22-03605-f002:**
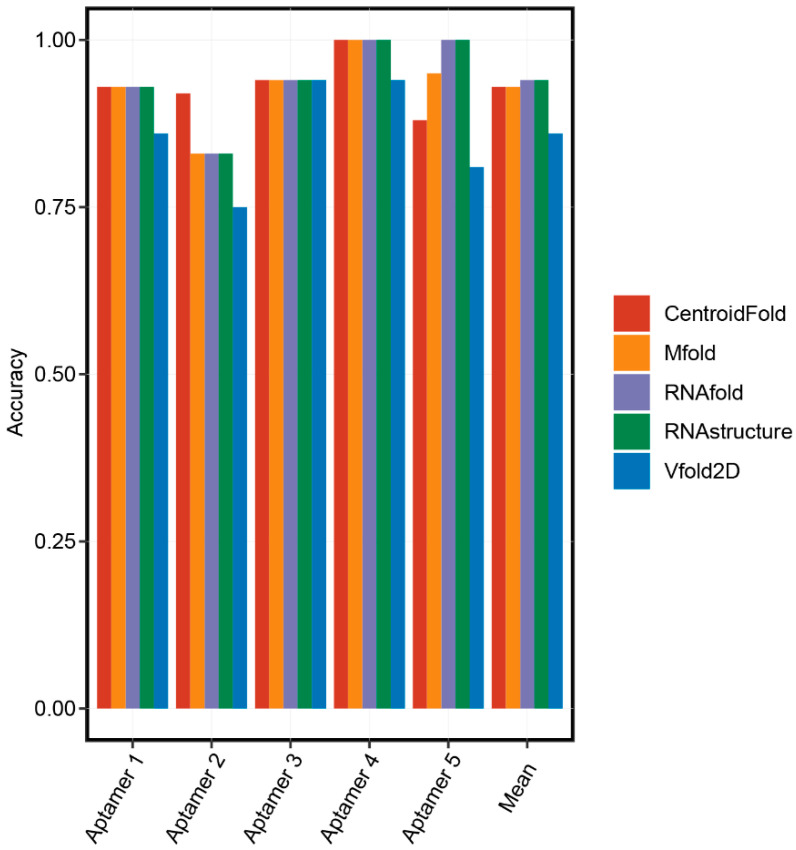
The accuracies of secondary structure prediction methods.

**Figure 3 ijms-22-03605-f003:**
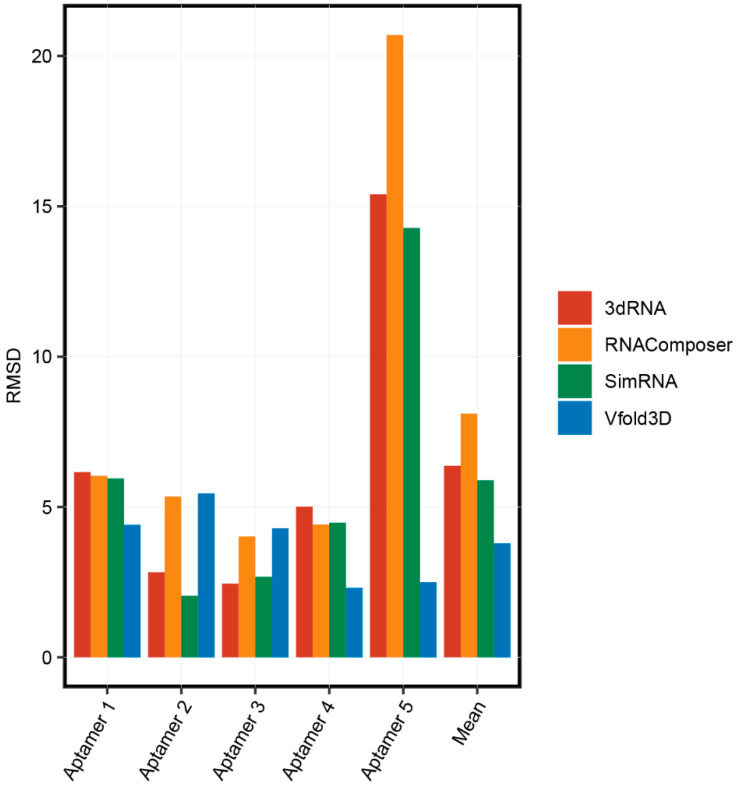
The accuracies of 3D structure prediction methods.

**Figure 4 ijms-22-03605-f004:**
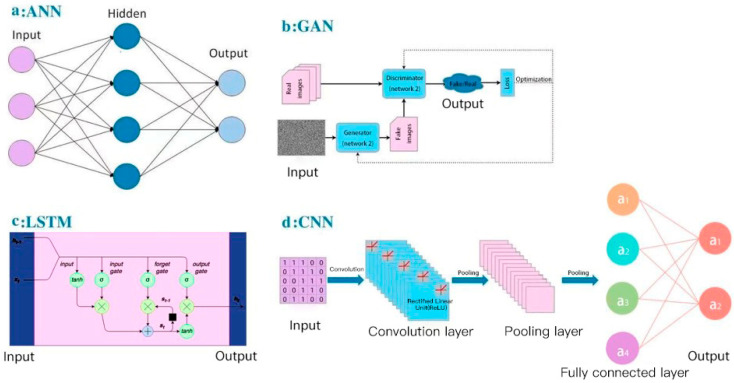
Diagrams of AI/deep-learning models in aptamer predictions: (**a**) artificial neural networks (ANN); (**b**) generative adversarial networks (GAN); (**c**) long short-term memory networks (LSTM); (**d**) convolutional neural networks (CNN).

**Table 1 ijms-22-03605-t001:** Methods for aptamer secondary structure prediction.

Software	Website Address	Developers	Example
RNAfold	http://rna.tbi.univie.ac.at/cgi-bin/RNAWebSuite/RNAfold.cgi (accessed on 4 March 2021)	Energy minimization [13]	RNAfold was selected to predict the tetracycline aptamer [21]
Mfold	http://www.unafold.org/ (accessed on 4 March 2021)	Energy minimization [22]	Four ssDNA aptamers were selected to inhibit the activity of angiotensin II [23]
RNAstructure	https://rna.urmc.rochester.edu/RNAstructure.html (accessed on 4 March 2021)	Energy minimization [24]	DNA aptamers against 17β-estradiol and the secondary structures of the aptamers were predicted using RNAstructure [25]
Vfold2D	http://rna.physics.missouri.edu/vfold2D/ (accessed on 4 March 2021)	Energy minimization [26]	The secondary structures of aptamers against human immunodeficiency virus-1 reverse transcriptase (HIV-1 RT) were predicted from the sequence by using the Vfold2D program [27]
CentroidFold	http://rtools.cbrc.jp/centroidfold/ (accessed on 4 March 2021)	Homologous sequence information [28]	The CentroidFold web server was used to predict the secondary structures of RNA aptamers targeting angiopoietin-2 [29]

**Table 2 ijms-22-03605-t002:** The aptamers selected for accuracy validation of computational tools.

No.	Aptamer	Sequence	PDB ID	Structure
1	RNA aptamer for *Bacillus anthracis* ribosomal protein S8	GGGCAGUGAUGCUUCGGCAUAUCAGCCC	2LUN	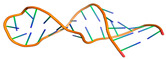
2	RNA aptamer for human IgG1	GGAGGUGCUCCGAAAGGAACUCCA	3AGV	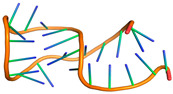
3	RNA aptamer for human immunodeficiency virus type-1 (HIV-1) reverse transcriptase	UACCCCCCCUUCGGUGCUUUGCAC CGAAGGGGGGG	6BHJ	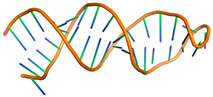
4	RNA aptamer for HIV-1 Rev protein	GGCUGGACUCGUACUUCGGUACUG GAGAAACAGCC	6CF2	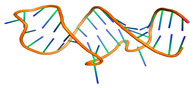
5	RNA aptamer for antibody fragments	GACGCGACCGAAAUGGUGAAGGACG GGUCCAGUGCGAAACACGCACUGUUG AGUAGAGUGUGAGCUCCGUAACUGGUCGCGUC	6B14	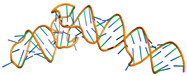

**Table 3 ijms-22-03605-t003:** Online web servers for the RNA aptamer 3D structure prediction.

Software	Website Address	Developers	Example
RNAComposer	http://rnacomposer.cs.put.poznan.pl/ (accessed on 4 March 2021)	Secondary structure elements [38]	RNA aptamers targeting angiopoietin-2 [29]
3dRNA	http://biophy.hust.edu.cn/3dRNA (accessed on 4 March 2021)	Secondary structure elements [39]	RNA aptamer targeting *Streptococcus agalactiae* surface protein [40]
Vfold3D	http://rna.physics.missouri.edu/vfold3D/ (accessed on 4 March 2021)	Secondary structure elements [26]	RNA aptamer targeting prostate-specific membrane antigen [41]
simRNA	https://genesilico.pl/SimRNAweb (accessed on 4 March 2021)	Lowest free energy [42]	RNA aptamers targeting angiopoietin-2 [43]

**Table 4 ijms-22-03605-t004:** Variation of binding energies between the predicted structures and the determined structures of aptamers in docking with the target protein.

Methods	Energy_Mixed	Energy_Protein	Energy_Aptamer	Energy_Binding	Energy_Binding_Variation
Reference	−13,472	−7983	−3318	−2171	0
Vfold3D	−13,324	−7983	−3661	−1680	491
SimRNA	−13,112	−8232	−3852	−1028	1143
RNAcomposer	−12,971	−8117	−3832	−1022	1149
3dRNA	−13,229	−8159	−3834	−1236	935

## Data Availability

No new data were created or analyzed in this study. Data sharing is not applicable to this article.

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
