# Peer review of "Artificial Intelligence in Aptamer–Target Binding Prediction"

_ijms, 2021, doi:10.3390/ijms22073605_

Round 1

Reviewer 1 Report

In this review, Chen and co-workers summarize several studies dedicated to in silico aptamer design. The topic of this review is timely to the field and tightly focused. However, the overall writing framework of this manuscript is very similar to that of the previous review article from last November [1], even the Figure ( “Typical workflow of aptamer in silico design and analysis”) is almost identical.

Most of the concepts mentioned in this review are similar to those found in the previous review paper. After comparing these two review papers, the new information of this review is limited.

In my opinion, this paper did not give me the satisfaction of reading it because a similar review article had been published in the same journal a few months ago, so I do not recommend it for publication.

Reference

[1] Methods and Applications of In Silico Aptamer Design and Modeling Int. J. Mol. Sci. 2020, 21(22), 8420; https://doi.org/10.3390/ijms21228420

Reviewer 2 Report

Authors provided a complete overview of methods for prediction of secondary and tertiary structures of RNA and DNA aptamers. The importance of aptamer structures in target recognition has been also fully considered and argued, as well as the novel approaches based on artificial intelligence aimed at predicting the aptamer-target binding. Review is well-written and clear, but some important points are missing or misleading. Thus, I suggest the following revisions. i) G-quadruplexes, representing main aptamer structures, have been overlooked. Recent articles and reviews, concerning their structure prediction as well as the importance of their fold in target recognition, should be cited and briefly discussed. See among others: 1) A guide to computational methods for G-quadruplex prediction; 2) The role of G-quadruplex structures of LIGS-generated aptamers R1.2 and R1.3 in IgM specific recognition; 3) G-quadruplex-based aptamers targeting human thrombin: Discovery, chemical modifications and antithrombotic effects; 4) Molecular dynamics simulations of human α-thrombin in different structural contexts: evidence for an aptamer-guided cooperation between the two exosites. ii) The sentence ‘A special case of Watson-Crick base pair’ (lines 125-126) associated to G-quadruplexes is misleading. Indeed, Hoogsteen base pairs are far more relevant in G-quadruplex structures than Watson-Crick base pairs.

Reviewer 3 Report

The manuscript «Artificial Intelligence in aptamer-target binding prediction» by Chen et al. concerns an actual problem of how to predict the properties of aptamers for their further use in practice. This problem is very important, since the development costs of aptamers against various targets will be minimized after this problem will be solved.

The review by Chen et al. is written in much detail — particularly the section, where 3D structures of aptamers are considered. Various methods, employed for the prediction of secondary and tertiary structure of aptamers, are compared. The aspects of the use of artificial intelligence for the prediction of probability of aptamer/target binding are discussed.

The manuscript is written in a good language. I recommend this manuscript for publication after just several minor points will be improved in order to enhance the manuscript quality. Below, these minor points are listed.

Point 1. Concrete values of affinity of the predicted aptamers against their targets, and the order of their variation upon the use of different software are not listed, while this is very important for further practical use of these aptamers.

Point 2. I recommend to use graphical block diagrams of the most efficient algorithms in order to illustrate the discussion on the use of learning technologies employed in (artificial intelligence)-based approaches. This will help readers to better understand what allows for the advantages of these algorithms.

Point 3. The authors are encouraged to specify the optimal length of aptamers used as protein-binding analogues of antibodies.

Round 2

Reviewer 1 Report

I have no further comments.